# Fungicide Activity of Culture Extract from *Kocuria palustris* 19C38A1 against *Fusarium oxysporum*

**DOI:** 10.3390/jof8030280

**Published:** 2022-03-09

**Authors:** Andi Setiawan, Fendi Setiawan, Ni Luh Gede Ratna Juliasih, Widyastuti Widyastuti, Aspita Laila, Wawan A. Setiawan, Fernandy M. Djailani, Mulyono Mulyono, John Hendri, Masayoshi Arai

**Affiliations:** 1Department of Chemistry, Faculty of Mathematics and Natural Science, Lampung University, Bandar Lampung 35145, Indonesia; andi.setiawan@fmipa.unila.ac.id (A.S.); fdsetiawan05@gmail.com (F.S.); niluhratna.juliasih@fmipa.unila.ac.id (N.L.G.R.J.); widyastuti.unila@gmail.com (W.W.); aspita.laila@fmipa.unila.ac.id (A.L.); mulyono@fmipa.unila.ac.id (M.M.); 2Department of Biology, Faculty of Mathematics and Natural Science, Lampung University, Bandar Lampung 35145, Indonesia; wawan.as@fmipa.unila.ac.id; 3Department of Fish Processing, Faculty of Fisheries and Marine Science, Gorontalo State University, Gorontalo 96128, Indonesia; fernandydjailani@ung.ac.id; 4Graduate School of Pharmaceutical Sciences, Osaka University, 1-6 Yamada-oka, Suita, Osaka 565-0871, Japan

**Keywords:** marine-derived actinomycetes, antifungal, *Fusarium oxysporum*, colloidal chitin

## Abstract

Secondary metabolites of actinomycetes are a potential source of bioactive compounds in the agricultural sector. This study aimed to determine the fungicidal properties of extracts of marine organism-derived actinomycetes. Actinomycetes were isolated from marine organisms using agar media with 1% colloidal chitin in artificial seawater. Then, the isolates were cultured on liquid media with 1% colloidal chitin in artificial seawater under static conditions for 14 days. The culture was extracted, the fungicide properties were evaluated using the microtiter 96-well plate method, and the influence of inhibition was visualized using apotome and SEM. Finally, the active extract was analyzed using LCMSMS. In the present study, 19 actinomycetes were isolated from marine organisms, and the isolates were examined with regard to their antifungal activities. Of these nineteen isolates, the isolate 19C38A1 was picked out from the rest. Hence, it showed significant control towards *F. oxysporum*. The prospective strain 19C38A1 was determined to be *Kocuria palustris* 19C38A1. The extract 19C38A1 was shown to cause damage to cell integrity, indicated by the shrinking form, and inhibited germination in the *F. oxysporum*; subsequently, the chemical characteristics of the compound produced by the potential isolate 19C38A1 indicated the presence of benzimidazole compounds in the active fraction of C38BK2FA. These results indicate that actinomycetes derived from marine organisms near the coast of Oluhuta, Tomini Bay, Gorontalo, related to strain 19C38A1, are not widely known as sources of valuable fungicides. This preliminary information is important, as it can be used as a basis for further development in the search for fungicides derived from marine actinomycetes.

## 1. Introduction

Losses of agricultural products are generally caused by crop diseases, most of which are caused by soil-borne fungi, *Fusarium oxysporum*. This fungus is the main cause of the decline in the productivity of agricultural commodities such as bananas and cereal-based crops [1]. This is because plants are damaged by this fungus, and in addition, the rate of the spread of the fungus is relatively high [2]. Currently, synthetic fungicides are still used to overcome this obstacle. The application of pesticides increased by 3.5 million tons by 2020 [3]. However, the intensive use of synthetic fungicides has an impact on environmental pollution and the spread of plant disease outbreaks [4]. Given the risks posed by synthetic fungicides and the increasing global preference for safer and environmentally friendly alternatives, new and valuable fungicides must be developed [5].

Based on a literature review, azole groups were found to be one of the active compounds commonly found in synthetic fungicides [6]. The active azole moiety targets the cytochrome P450 enzyme, known as sterol 14α-demethylase (CYP51) or lanosterol 14α-demethylase, and inhibits the initial step in ergosterol biosynthesis [7]. More specifically, the benzimidazole group has broad-spectrum activity; it interacts with β-tubulin and inhibits hyphal growth [8]. The benzimidazole group is a strong inhibitor of tubulin polymerization and exerts antifungal activity by targeting the β-tubulin subunit of microtubules, which means cell division is not carried out, leading to cell death.

In recent years, some researchers have looked for antifungal compounds derived from microorganisms. Actinomycetes are potential sources of active metabolites with very diverse structures and different activities [9]. The results of a recent study reported that *Streptomyces* sp. SY195 is able to produce antifungal compounds that inhibit the growth of *Candida albicans* in Gauze liquid media [10]. It has also been reported that *Streptomyces sporoclivaturs* NBRC 100,767 cultivated on ISP3 media was able to produce fungicidal compounds that could inhibit the growth of *F. oxysporum* [11]. However, to date, information regarding azole fungicides, especially those produced from rare actinomycetes cultivated on liquid chitin colloid media, is still scarce.

In this study, we aimed to isolate actinomycetes from marine organisms and to screen, purify, and characterize culture extracts that display antifungal activities. A selected isolate was chosen for the further analysis of fungicide activity. In addition, shrimp shell waste was used as a source of colloidal chitin, which would be used as main a carbon source in actinomycetes growth media.

## 2. Materials and Methods

### 2.1. Biomaterials

#### 2.1.1. Preparation Shrimp Shell Waste and Colloid Chitin

Shrimp shell waste was collected from free-market Gudang Lelang, Teluk Betung Bandar Lampung. The shrimp shells were washed with tap water to remove impurities such as dust and soil. The shrimp shells were then dried overnight in an oven at 60 °C. The dried shrimp shells were demineralized using HCl 2 N, stirred at 30 °C for 2 h, and then neutralized with distilled water. After that, the dried, demineralized residue was deproteinated with NaOH 2.5%, stirred at 90 °C for 2 h, and then neutralized using distilled water until a pH of 7 was reached. The pellet was dried in an oven at 60 °C overnight to obtain the chitin flake. Chitin was transformed into colloidal chitin by concentrated HCl being added and it being stirred for 2 h; then, it was neutralized with distilled water to a pH of 7 and stored in a freezer at −20 °C until it was used in further experiments [12].

#### 2.1.2. Actinomycetes Isolates

Samples of marine organisms were collected from Tomini Bay, Gorontalo, Indonesia (0°25′11.9″ N 123°08′31.8″ E), in August 2019. Small pieces of marine organisms were rinsed with sterile artificial seawater (ASW). The pieces were placed into plates containing colloidal chitin agar medium supplemented with 25 μg/mL of cycloheximide and 25 μg/mL of nalidixic acid, and they were cultured at room temperature for 14 days. The isolation of actinomycetes was carried out via the dilution method using agar media, 1% colloid chitin, and ASW (gL^−1^) consisting of 27.0 g of NaCl, 5.6 g of MgCl_2_·6H_2_O, 1.5 g of CaCl_2_·2H_2_O, 1.0 g of KNO_3_, 0.07 g of K_2_HPO_4_, 6.6 g of MgSO_4_·7H_2_O and 0.04 of NaHCO_3_ [13,14,15].

#### 2.1.3. Pathogenic Fungi

The phytopathogen *F. oxysporum* used in this study was collected from the Microbiology Laboratory, Department of Biology, Lampung University. *Malassezia Globosa* were collected from the Department of Parasitology, University of Indonesia. Both fungi were maintained in potato dextrose agar (PDA) media.

### 2.2. Cultivation

The inoculum of actinomycetes was prepared using 100 mL of liquid colloidal chitin containing 1% in artificial seawater (ASW) in a 500 mL culture flask, which was incubated for 7 days under static conditions. Seven-day-old inoculum was transferred into a liquid fermentation medium containing 500 mL of 1% colloidal chitin in ASW in a 2000 mL Erlenmeyer flask. The culture was incubated for 14 days at room temperature under static conditions.

### 2.3. Screening of Antifungal Activity

The high-throughput screening method was used to determine the antifungal activity using 96-well plates, and absorbance was measured using a Hospitex plate reader at OD_630_. Seven-day-old fungi were cultured on potato dextrose agar (PDA). Spores were harvested using a moistened sterile cotton swab; before being used, the cotton swab was moistened in a tube containing a drop of Tween 80 in potato dextrose broth (PDB) and transferred into a 10 mL inoculum tube containing PDB. The tube was vortexed for 15 s. The heavy particles were allowed to settle down for 3–5 min. The upper homogenous suspension was separated from heavy particles into a new, sterile tube. Inoculum was adjusted to 0.5 Mc Farland standard turbidity equal to (10^6^ CFU/mL) or (OD_630_ = 0.08–0.1) [16]. All *n*-butanol extracts were made for 10 mg/mL of stock solution. Ketoconazole was used as a positive control, 12.5% MeOH was used as a solvent control, and wells without fungus were used as a control of contamination. Each well contained 80 μL of PDB media, 100 μL of fungal suspension at 1.0 × 10^5^ CFU (colony-forming units)/mL, and 20 μL of extract solution [17]. The plate was incubated for 48 h at room temperature. The % of inhibition was measured as follows:$$\%\;\mathrm{of}\;\mathrm{Inhibition}=\frac{\mathrm{OD}\;\mathrm{Control}-\mathrm{OD}\;\mathrm{Treatment}}{\mathrm{OD}\;\mathrm{Control}}\times 100\%$$

### 2.4. Characterization of Selected Actinomycetes

#### 2.4.1. Morphological Study

The arrangement of spore ornament structure was examined using cover slip culture medium and observed in 400 M microscopically [18]. A clean sterile coverslip was planted at a tan angle of 45° in an agar plate and was cultured for 7 days. The coverslip was removed from the agar plate using a tweezer and placed upwards on a clean glass slide. The coverslip was observed under the Zeiss Axio Imager Microscope (400×). A scanning electron microscope (SEM) study was performed based on mycelium and spore ornament identification. The inoculum of selected actinomycetes was prepared in a 1% colloidal chitin liquid medium in artificial seawater and incubated under static conditions. After 7 days, the shrimp shell was placed onto a clean Petri dish. One milliliter of actinomycetes suspension was added to moisten the one gram of shrimp shell. Then, the culture was incubated for 7 days under static conditions. After the incubation, a small piece of shrimp shell was cut using the SLEE Disposable Blades microtome. The sample was prepared by it placing on aluminum stubs, which were fixed with carbon adhesive tabs. Then, gold plating was completed in 20 min. The gold-plated metal stub was observed using SEM EVO with 10 kV electron high voltage, Carl Zeiss EVO MA 10, Oberkochen, Germany.

#### 2.4.2. Phylogenetic Analysis of Strain 19C38A1

Genomic DNA was extracted following the genomic Wizard^®^ Genomic DNA KIT protocol (cat. No. A1120, Promega, Madison, WI, USA). PCRs of 16S rDNA sequences were completed using a Sensoquest Sensodirect thermocycler from Germany. PCRs were performed using a forward primer: 5′-AGA GTT TGA TCM TGG CTC AG-3′ [19]. Additionally, a reverse primer: 5′-CCG TAC TCC CCA GGC GGG G-3′ [20], which amplified 810 bp, was used. The PCR reactions were completed using a 2G Fast ReadyMix Kit (cat. No. KK5102, Merck, Taufkirchen, Germany). The PCR reactions were carried out at a total volume of 25 µL, containing 5 µL of DNA template (50 ng/L), 12.5 µL of 2G Fast ReadyMix, 6.5 µL of RNAse-free water, 0.5 µL of forward primer, and 0.5 µL of the reverse primer. Amplification was carried out in 35 cycles as follows: denaturation for 60 s at 92 °C, primer annealing for 60 s at 54 °C, and polymerization for 90 s at 72 °C. Bends were detected with the Qiaxcel Advanced instrument (Qiagen, Hilden, Germany) according to the protocol, then sequenced using the Sanger method. The results of the sequencing were analyzed phylogenetically using Mega version 11 software. 

### 2.5. Scale-Up Cultivation and Extraction

The selected isolate 19C38A1 was cultured (twice) in 2 L of 1% colloidal chitin liquid medium for 14 days under static conditions, and the biomass was harvested using a centrifuge at 7000 rpm, 4 °C, for 10 min to separate the supernatant and pellet. The filtrate was partitioned using *n*-butanol, and the organic fraction was concentrated via evaporation at 40 °C under reduced pressure. The organic fraction was fractionated using silica gel 60 (0.063–0.200 mm, Merck KgaA, Darmstadt, Germany) open-column chromatography using *n*-hexane and EtOAc. Thin-layer chromatography (TLC) was conducted using silica gel F_254_ (aluminium sheets, Merck KgaA, Darmstadt, Germany) and specific reagent Dragendorff’s (Solution 1: 1.7 g of bismuth nitrate with 80 mL of water and 20 mL of glacial acetic acid; Solution 2: KI solution (50% *w/v*, 100 mL) with glacial acetic acid), and ninhydrin (2% *w/v* in 10 mL of ethanol).

### 2.6. Characterization

#### 2.6.1. Microscopy Analysis of Fungicide Activity

To verify the inhibitory ability of the potent isolate extract against *F. oxysporum*, further microscopy analysis was carried out using an Apotome Microscope and SEM EVO Carl Zeiss EVO MA 10, Oberkochen, Germany. At first, spore germination was observed in the *F. oxysporum* population, which occurred by transferring 20 μL of the 48 h incubation results on a glass slide. Samples were observed at 400 M magnification. Conidia were considered germinated if the length of the germ tube was wider than the conidia. Further analysis used SEM to visualize the surface morphology of the hyphae of *F. oxysporum* and to determine the possible mechanism between the crude extract of potent isolate and *F. oxysporum*. For microscopic analysis, the 96-well plates used were modified with carbon tape. The wells were given a mixture of 100 μL of extract and 100 μL of *F. oxysporum* (10^6^ spores/mL). *F. oxysporum* (10^6^ spores/mL) was used as a control. At the edge of the well, sterile carbon tape was attached. The plate was then incubated for 48 h at 30 °C. In the step prior to SEM analysis, the carbon band with adhering hyphae was carefully removed and washed three times with 0.1 M PBS [21]. Samples were fixed with 10% (*v*/*v*) formaldehyde overnight at room temperature, then rinsed three times with 0.1 M PBS and dehydrated in graded ethanol (30% for 10 min, 50% for 10 min, 70% for 10 min, 90% for 10 min, and 100% for 1 h). Samples were air dried, mounted on aluminum stubs using double-sided carbon tape, sputtered with gold, and visualized using SEM EVO Carl Zeiss EVO MA 10, Oberkochen, Germany.

#### 2.6.2. Liquid Chromatography–Mass Spectrometry (LC-MS)

The identification and characterization of fungicide compounds produced by isolate 19C38A1 were carried out using LC-MSMS Positive Mode. The active fraction was dissolved in methanol and analyzed via LC-MSMS analysis, which was equipped with the ACQUITY UPLC^®^ H-Class System (Waters, Beverly, MA, USA), ACQUITY UPLC^®^ HSS C18 column (1.8 μm 2.1 × 100 mm) (Waters, Beverly, MA, USA), and Xevo G2-S Qtof Mass Spectro (Waters, Beverly, MA, USA).

## 3. Results and Discussion

### 3.1. Sample Collection and Isolation of Actinomycetes

Actinomycetes were isolated from 17 marine organisms collected from near the coast of Oluhuta, Tomini Bay, Gorontalo, Indonesia (0°25′11.9″ N 123°08′31.8″ E). Samples of marine organisms were collected via Self-Contained Underwater Breathing Apparatus (SCUBA) diving at depths of 5–20 m. Tomini Bay waters are located on the equator covering three provinces, namely North Sulawesi, Gorontalo, and Central Sulawesi. Tomini has high marine biodiversity, evidenced by the variety of fish species, coral reefs, seagrass beds, and mangroves. Tomini Bay is the second-largest bay in Indonesia after Cendrawasih Bay. Tomini geologically formed around six million years ago and has an area of 70.020 km^2^. Tomini is rich in geological history, and this small region has a very high level of endemism [22].

In this study, 19 actinomycetes associated with marine organisms were isolated from 15 sponges, 1 tunicate, and 1 macroalga. The presence of actinomycetes in marine environments has been reported to be capable of producing bioactive metabolites with diverse activities [23]. Furthermore, the presence of actinomycetes has been found in a range of marine organisms such as sponges [24], tunicates [25], and macroalgae [26].

### 3.2. Screening of Antifungal Activity

The results of the bioactivity test of the actinomycetes extract against *F. oxysporum* and *M. Globosa* are shown in Table 1. Almost all the extracts had inhibitory activity levels below 50%, except for isolate 19C38A1 from the sponge, which had an inhibitory activity level of 54%.

Several studies have reported that sponge-derived actinomycetes can produce secondary metabolite compounds with a variety of therapeutic activities; isolates such as *Gordonia*, *Kocuria*, *Nocardia*, *Micrococcus*, *Micromonospora*, and *Microbacterium* have been successfully isolated from marine sponges [27]. Moreover, tunicate was also reported to be a host for several types of actinomycetes such as genus of *Actinomadura*, *Aeromicrobium*, *Arthobacter*, *Brevibacterium*, *Curtobacterium*, *Gordonia*, *Kocuria*, *Micrococcus*, *Micromonospora*, *Nocardia*, *Nocardiopsis*, *Saccharopolyspora*, *Salinispora*, *Solwaraspora,* and *Verrucosispora* [28]. However, extract isolates 19B19A1 and 19B19A2 from tunicate showed relatively weak activity. The same weak activity was observed in extract of isolate 19D40A3 from macroalgae. In addition, the presence of actinomycetes associated with macroalgae such as *Rhodococcus, Nonomuraeae, Microbispora, Isoptericola, Nocardiopsis*, and *Microbacterium* have been reported [29]. Based on the screening data, it appears that actinomycete 19C38A1 has the potential to produce fungicides. Therefore, isolate 19C38A1 was selected for further research related to understanding its morphology and phylogenetic properties, as well as characterization of the metabolites produced.

### 3.3. Characterization of Selected Actinomycetes

#### 3.3.1. Morphological Study

The morphology of the isolated 19C38A1 grown for 7 days on 1% colloidal chitin media is shown in Figure 1a,b. In addition, the surface of the 19C38A1 isolate on the shrimp shell was very clearly observed, as shown in Figure 1c. The results of observations (Figure 1a) showed white colonies with aerial mycelium, in accordance with the general characteristics of actinomycetes [30]. Further observation using a light microscope at 400× magnification (M) showed the presence of spore fragmentation, which characterizes the family Micrococcaceae [31]. Advanced analysis using SEM enabled us to observe areal mycelia that displayed a straight spore chain with a diameter of 2 microns (Figure 1c) [32].

#### 3.3.2. Phylogenetic Analysis of Isolates 19C38A1

Sequencing of the 16S rDNA gene indicated the genus Kocuria. The new isolate, *Kocuria palustris* 19C38A1, was identified via sequencing with a similarity percentage of 99.77% and was successfully registered in GenBank with access number LC659429. Based on phylogenetic analysis, it is known that this isolate is known as *K. palustris* (Figure 2). According to the literature, *K. palustris* belongs to a rare group of actinomycetes that have been obtained from marine habitats [33]. This group of rare actinomycetes actually reflects a potential source of active metabolites that have not been studied much in terms of the structure and properties of their activities. Moreover, the presence of Micrococcaceae is common in terrestrial and marine sources. Although Micrococcaceae are widely distributed in sponges, there is very little information regarding natural products of bioactive compounds, especially from the genus Kocuria, associated with sponges [34].

### 3.4. Scale-Up Cultivation and Extraction

To produce the biomass of actinomycetes, isolate 19C38A1 was cultured in liquid fermentation in two 2 L Erlenmeyer flasks. After 2 weeks of cultivation, actinomycetes were formed in surface media, but they were clumsy, had an insoluble structure, and were not pigmented. The culture media was centrifuged to separate the solids and the filtrate. Then, the filtrate was partitioned with *n*-butanol. The *n*-butanol extract of 19C38A1 was evaporated under reduced pressure to afforded crude C38B extract (1.2 gr). The C38B extract was subjected to several steps of open-column chromatography using SiO_2_ as a stationary phase, and it was eluted with *n*-hexane: EtOAc (linier gradient) to obtain a 4 mg active fraction of C38BK2FA. The results of the TLC analysis, as shown in Figure 3a, confirm the presence of orange spots from Dragendorff’s specific reagent, indicating the presence of alkaloid. The same was observed for the orange spots on the C38BK2FA fraction, as shown in Figure 3b, with a higher orange color intensity and an elongated shape. This indicates the presence of several alkaloid components in the C38BK2FA fraction. Further tests on the inhibitory activity showed that the C38BK2FA fraction had 88% inhibition against the growth of *F. oxysporum* at a concentration of 1 mg/mL.

### 3.5. Characterization

#### 3.5.1. Microscopic Analysis of Fungicide Activity

Analysis based on observations using apotome microscopy showed clear differences in fungal cell morphology between the control and *F. oxysporum* treated with C38BK2FA extract. It was seen that in the control, *F. oxysporum* grew regularly, with elongated and branching hyphae (Figure 4a). In the treatment of the C38BK2FA extract sample, it was seen that there was prominent inhibition; the compound extract had an effect on mycelial growth and conidia germination, which was characterized by the absence of apical cell elongation (Figure 4b). It was found that the bioactive compounds present in the C38BK2FA extract created a stressful environment that inhibited conidia germination and hyphal growth. Further observations using SEM enabled us to obtain an overview of the effect of damage on the morphology of hyphae and conidia. In the control, it was seen that the hyphae growth was regular, which displayed a tubular structure, flat width, smooth surface, and elongated shape. The conidia showed a characteristic cube shape and a flat bottom (Figure 4c). On the other hand, after being treated with the addition of C38BK2FA extract, the hyphae structure was wrinkled and distorted (Figure 4d). Changes in the structure of hyphae and conidia were correlated with the assay for the inhibition of conidia germination. Our observations of this fungicidal effect are similar to those of Koch and Loffer [35], who observed the effect of actinomycete bioactive compounds on phytopathogens.

#### 3.5.2. Liquid Chromatography–Mass Spectrometry

The results of the structural analysis of the active C38BK2FA fraction using ESI-TOF-MS showed the presence of 10 chromatogram peaks, as shown in Figure 5. Meanwhile, the interpretation of the mass spectroscopic data showed that the four chromatogram peaks at retention times 13.65 (1), 14.00 (2), 14.4 (3), and 15.80 (4) minutes had molecular peak ions [M + H]^+^ at *m*/*z* 367.2429, 365,2344, and *m*/*z* 367.2344, with the molecular formula C_22_H_28_N_4_O, and the molecular peak ion [M + H]^+^ at *m*/*z* 447.3065 with the molecular formula C_23_H_38_N_6_O_3_, respectively. The data show that the active C38BK2FA fraction had nitrogen content, and this was confirmed by the data from the previous TLC test results. Based on the database on Chemspider, the molecular formula C_22_H_28_N_4_O is an alkaloid group of benzimidazole with the name N-(4-Methoxybenzyl)-1-[2-1-piperidinyl) ethyl]-1H-benzimidazole-2-amine. Meanwhile, the compound with the molecular formula C_23_H_38_N_6_O_3_ is a triazole compound with the name Methyl 1-[(3R,5S)-5-(cycloheptylcarbamoyl)-1-(1-ethyl-4-piperidinyl)-3-pyrrolidinyl]-1H-1,2,3-triazole-4-carboxylate. Based on the data obtained, it can be concluded that the fungicidal activity of the active C38BK2FA fraction is due to the presence of benzimidazole compounds.

Compounds in the benzimidazole group have unique amphoteric properties and contain two nitrogen atoms. This means this group of compounds can be acidic and basic. Moreover, the benzimidazole ring compound has two tautomeric forms, in which a hydrogen atom is located on two nitrogen atoms. Then, electron-rich nitrogen heterocycles can accept or donate protons and easily form various weak interactions. Compounds in the benzimidazole group easily bind to therapeutic targets, which makes compounds in the benzimidazole group broad-spectrum agents [36]. Benzimidazole derivates play an important role in controlling various fungal pathogens [37].

Benzimidazole belongs to a group of broad-spectrum systemic fungicides that have been used to control plant diseases caused by *Fusarium* spp. by interacting with tubulin and inhibiting hyphae growth. The mechanism of action is related to the inhibition of fungal growth through impaired tubulin polymerization [38]. In addition, the presence of azole group compounds triggers fluidity, asymmetry, and integrity in the fungal cell membrane [39]. Triazole group compounds are also known to be active against most Candida species. However, this group of compounds did not display activity on most fungi such as *Aspergillus* spp., *Fusarium* spp., and Mucormycetes [40].

## 4. Conclusions

These findings concluded that the C38BK2FA extract fraction of *Kocuria palustris* 19C38A1 displayed 88% inhibition of the growth rate of *F. oxysporum* at a concentration of 1mg/mL. This activity can be caused by the presence of components of the benzimidazole group of alkaloid extracts. This pioneering information is very important for use in further research related to the development of new biofungicides that are environmentally friendly and sustainable.

## Figures and Tables

**Figure 1 jof-08-00280-f001:**
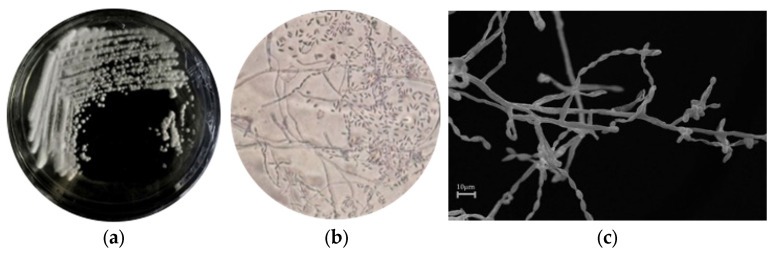
(**a**) Isolate 19C38A1 in colloidal chitin agar media 1%; (**b**) visualization of 19C38A1, with light microscopic scale 400×; (**c**) SEM image of aerial hyphae isolate 19C38A1, bar 2 μm.

**Figure 2 jof-08-00280-f002:**
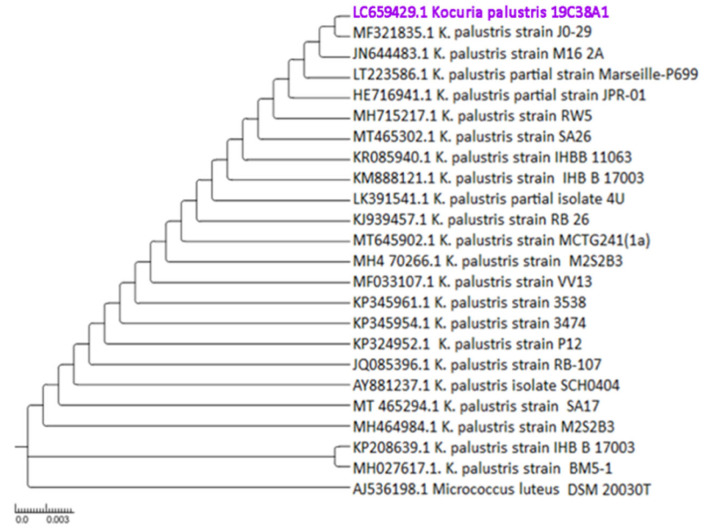
Phylogenetic tree using maximum likelihood method and Hasegawa–Kishino–Yano model of 23 Kocuria representatives and *Micrococcus luteus* (Acc. No. AJ536198) as an outgroup. Bootstrap values (1000 resamples) are given in percentages at the nodes of the tree. The isolate *Kocuria palustris* 19C38A1 was presented.

**Figure 3 jof-08-00280-f003:**
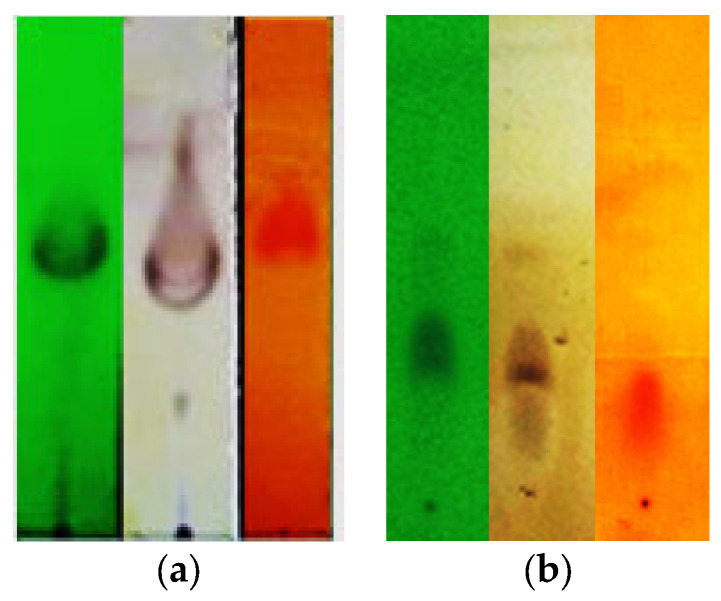
TLC result of (**a**) crude C38B extract with eluent *n*-hexane: EtOAc (1:1); (**b**) active C38BK2FA fraction with eluent *n*-hexane: DCM (10:1).

**Figure 4 jof-08-00280-f004:**
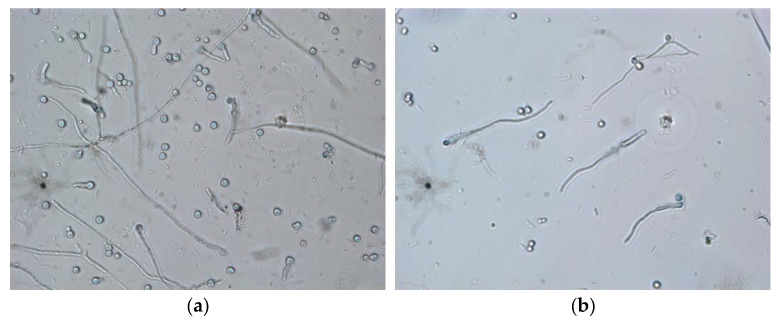
(**a**) Control of *F. oxysporum* in apotome microscopy observation 400 M; (**b**) *F. oxysporum* treated with crude C38BK2FA extract observed under apotome microscopy 400 M; (**c**) control of *F. oxysporum* observed using SEM 10 K x; (**d**) *F. oxysporum* treated with crude C38BK2FA extract observed using SEM 15 K x.

**Figure 5 jof-08-00280-f005:**
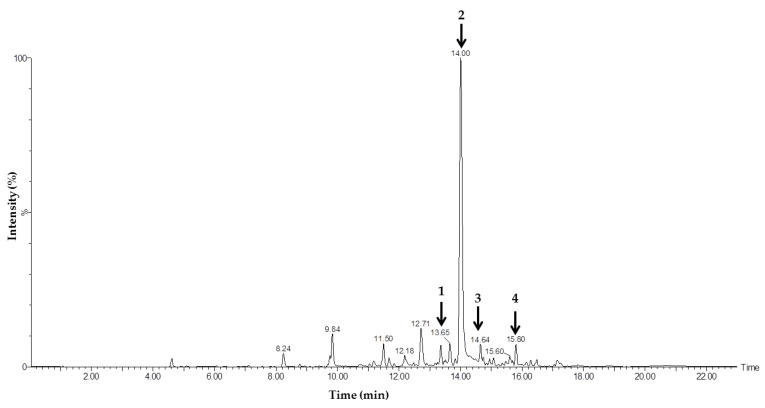
ESI TOF MSMS spectrum fraction C38BK2FA, [M + H]^+^ = *m*/*z* 365.2344.

**Table 1 jof-08-00280-t001:** Screening of antifungal assay.

No	Sample Code	Phylum	Isolate Actinomycetes	% Inhibition at 1.0 mg/mL	
				*F. oxysporum*	*M. globosa*
1	01A07	Porifera	19A07A1	0	5
2	01A13	Porifera	19A13A3	14	17
3	01A15	Porifera	19A15A1	0	3
4	01A18	Porifera	19A18A1	0	0
5	01B19	Tunicate	19B19A1	0	32
			19B19A2	3	8
6	01B20	Porifera	19B20A1	42	6
7	01B21	Porifera	19B21A1	1	36
8	02C30	Porifera	19C30A1	0	26
9	02C32	Porifera	19C32A1	0	31
10	02C33	Porifera	19C33A2	45	39
11	02C34	Porifera	19C34A1	0	32
12	02C35	Porifera	19C35A1	0	28
13	02C36	Porifera	19C36A1	0	46
14	02C38	Porifera	19C38A1	54	7
15	03D40	Macroalgae	19D40A3	0	29
16	03D41	Porifera	19D41A1	0	37
			19D41A2	0	13
17	03D46	Porifera	19D46A1	0	27

## Data Availability

Not applicable.

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
