# Peer review of "Fungicide Activity of Culture Extract from Kocuria palustris 19C38A1 against Fusarium oxysporum"

_jof, 2022, doi:10.3390/jof8030280_

Round 1

Reviewer 1 Report

In this work, the authors have studied that the extract fraction of Kocuria palustris 19C38A1 had an high inhibition of the growth rate of Fusarium oxysporum. The character of the activity could be caused by the presence of components of the benzimidazole group of alkaloid extracts. It is a good work but the introduction must be improved because there are some confuse sentences like this: "In line with the issues of globalization and trade,
this puts pressure on the rate of pathogen spread to increase and threatens agroecosystems". I accept it but the introduction must be improved

Author Response

 Dear Reviewers 1

We are grateful for helpful feedback from the Reviewers that helped us to improve the quality of the manuscript. We carefully responded to all points and have revised the manuscript accordingly. 

Reviewer 2 Report

Overview and general recommendation:

The manuscript by Setiawan and co-authors described the isolation of marine actinomycetes and the identification of a putative antifungal activity against F. fusarium. The authors described the isolation methods used for actinomycetes and the identification of the strains. The also showed a putative antifungal activity from fermented extract C38B from the isolate Kocuria palustris. This activity was described by the authors as a damage on the morphology of hyphae and conidia using microscopy techniques. The extract was also analysed by Liquid Chromatography- Mass Spectrometry and two molecules belonging to benzimidazole group were identified.

Major comments:

I suggest that the authors consider the following points in preparing the final manuscript:

  1. In the introduction, should be introduced more information about antifungal drugs such as benzimidazole, with a focus on their activities.
  2. The authors showed damage of fungus hyphae, that should be due not only to an antifungal drug but also to other compounds or enzymes. Did the authors performed inhibition assay on petri dishes? Did F. fusarium or other fungi were completely inhibited in their growth on solid or liquid media in the presence of the extracts?
  3. To conclude that the molecules identified with the LC-MS were responsible of the antifungal activity, the authors need to purified them and then used on F. fusarium or other fungi in order to verify the specific activity.
  4. The conclusions must be expanded with more literature data and with future prospective and applications.

Minor comments:

Please check the bacteria and fungi names (italics, spaces etc..)

Please improve quality/resolution of photos and figures

Author Response

 Dear Reviewer 2

We are grateful for helpful feedback from the Reviewers that helped us to improve the quality of the manuscript. We carefully responded to all points and have revised the manuscript accordingly 

Reviewer 3 Report

The work of Andi Setiawan and co-workers describes the characterization and antifungal activity of marine organisms-derived actinomycetes extracts.

The manuscript itself has many problems.

First of all, according to the Authors', section 2.3. Screening of Antifungal Activity was done according to the EUCAST guidelines. However, they did not follow EUCAST guidelines. Please check these guidelines carefully.

Generally, section 2. Materials and Methods is written not clear (e.g., section 2.2. Cultivation and 2.5. Cultivation and Extraction).

On what basis did the authors identify actinomycetes, only based on morphology? What were the colony and cell morphology of the 19C38A1 isolate? Figures 1a and 1b are not readable.

Incorrect numbering of the figures cited in the text (Fig. 4)

What was the inhibitory activity of the extract fraction of Kocuria palustris 19C38A1 against F.oxysporum, 54% or 88%?

In the present form, some data and their rationale are not clear.

Author Response

 Dear Reviewer 3, 

We are grateful for helpful feedback from the Reviewers that helped us to improve the quality of the manuscript. We carefully responded to all points and have revised the manuscript accordingly 

Round 2

Reviewer 2 Report

The authors followed the suggestions and improve the quality of the article.

Reviewer 3 Report

Thank you to the authors for taking into account my comments.